# Weak-to-Strong Confidence Prediction

## Abstract

As large language models (LLMs) are increasingly deployed across a wide range of application domains, understanding their capacity through uncertainty—especially in open-ended domains—is crucial to ensuring that they operate safely and reliably. Well-calibrated uncertainty estimates that accompany the text generated by an LLM can indicate the likelihood of an incorrect response, and as such, can serve as an effective fail-safe mechanism against hallucinations. Unfortunately, despite a growing body of research into uncertainty quantification in LLMs, existing methods largely fail to provide reliable uncertainty estimates in practice, and the lack of comparability across methods makes measuring progress difficult, necessitating the development of more robust methods that allow us to predict whether frontier models are able to provide a factual response to a given prompt. In this paper, we show that the probability of a frontier model providing a factually correct answer to a query can be predicted with high accuracy from smaller, weaker models. We believe that this work contributes to a deeper understanding of model capacity, particularly in terms of weak-to-strong generalization, and facilitates the creation of more trustworthy LLMs.

## 1 Introduction

Large Language Models (LLMs) are being increasingly used as vehicles for question answering and information retrieval in high-stakes scientific, business, and government settings. Because of this increase in usage it is paramount to user safety to develop models that do not deceive the user with their answers, a phenomenon known as hallucination (Xu et al., 2024). To mitigate this problem, we study whether *a second model* can oversee the output of a primary one: given a factual question and the answer provided by an LLM (the "generator") to that question, can *another LLM* (the "evaluator") tell us how likely is the answer to be right or wrong? We show that this is not only possible, but also that *the evaluator LLM can be orders of magnitude smaller than the question-answering LLM*.

This finding is crucial for the engineering of safe LLMs as it allows for several key takeaways:

1. The evaluator LLM can run locally on an end-user's machine, and can work even when the generator is a black-box model. This prevents potential tampering with the model, were it to be hosted on a remote server, like a larger LLM would have to be.

2. The evaluator LLM *does not need to know the answer* to the question in order to accurately judge whether the generator has answered it correctly. This suggests that the task of answering a question may be *intrinsically different* from the task of detecting the likely correctness of that answer.

3. The evaluator LLM can achieve *good calibration*. This is particularly important in our setting as the predicted probability of the correctness of an answer given by the generator LLM is our chosen measure of uncertainty.

Overall, these conclusions make contributions to the literature on uncertainty quantification in LLMs, in particular we add to the popular idea that LLMs can quantify their own uncertainty (Kadavath et al.,

Submitted to 38th Conference on Neural Information Processing Systems (NeurIPS 2024). Do not distribute.

2022) by showing that they can also quantify others. Our results show that harnessing uncertainty from frontier AI models may be a useful tools for interpreting ML models.

The paper will proceed by first introducing some background on uncertainty quantification in LLMs (Section 2), then introducing our experimental design, methods and dataset tested (Section 3), and finally by presenting and discussing our results (Section 4 and Section 5).

## 2   Background

**Uncertainty quantification in LLMs.**   When it comes to LLM research, model hallucination is one of the major concerns of that researchers hope to solve by quantifying model uncertainty (Yadkori et al., 2024). Many works develop novel metrics to compute uncertainty or confidence (Kuhn et al., 2023; Duan et al., 2024), while others understand uncertainty through narrowing down the range of data (Amayuelas et al., 2024; Yin et al., 2023). Another popular approach is to prompt LLMs to explicitly express their uncertainty (Lin et al., 2022; Tanneru et al., 2023). In this work, we will instead directly *learn* the probability that a model may be correct or incorrect about a specific question it is asked.

**Selective prediction.**   Besides knowing how capable a model is through accuracy, we also want to know if the evaluator is well-calibrated. This is crucial as this predicted probability is likely of more interest to an end-user than a simple yes/no judgement would be. To compute this, we introduce a rejection class (El-Yaniv and Wiener, 2010): if the evaluator's uncertainty is beyond a given threshold, the model abstains from making a prediction. By evaluating model outputs on a variety of thresholds, we can compute selective metrics like selective accuracy (Fisch et al., 2024). By comparing the selective metrics with the non-selective ones, we can better understand if our model is well-calibrated (Rudner et al., 2024; Varshney et al., 2022).

## 3   Experimental design

We employ a larger, more capable language model, denoted as the Generator $f_G$, to generate responses to questions from various datasets. In additon, We use a weaker model as the Evaluator $f_E$. Our goal is to train $f_E$ to learn the uncertainty of $f_G$ from the responses given the dataset $\mathcal{D}$. Below We describe our dataset construction and the design of $f_E$.

### 3.1   Dataset construction

For a given dataset $\mathcal{D}$, we collect $n$ questions from $\mathcal{D}$ and query $f_G$ to generate $k$ answers per question.[1] We then compare each generated answer to the true answer in $\mathcal{D}$ and obtain the labels $Y = \{1, 0\}$ indicating whether $f_G$ answers correctly. By averaging over the $k$ answers of each question, we compute a probability label $y$ of the answer from $f_G$ being correct.

We evaluate from two datasets: TriviaQA (Joshi et al., 2017) and MMLU (Hendrycks et al., 2021), as both contain a large corpus of questions and factual answers. For the open-ended TriviaQA, only the original question is presented to $f_G$. For the multiple-choice MMLU, we present both the question and the four choices together as a prompt to the $f_G$.

### 3.2   Evaluator training and evaluation setup

To train the Evaluator $f_E$, we leverage the high-dimensional representations of the input questions produced by a "backbone" LLM. These representations are inputs to a linear classification head. We experiment with two variants of this approach, a probe head setup and a supervised finetuning setup.

**Probe head classifier setup.**   We implement a probe head as a two-layer neural network that takes as input the high-dimensional text representations generated by an open-weight fixed "backbone" LLM. Following Kadavath et al. (2022), we use only the representation of the last non-padding token of each prompt, resulting in an input of shape $X \in \mathbb{R}^{n \times d}$, where $d$ is the output dimension of the final layer of $f_E$ before the linear head. Since $y$ represents a probability, the output of the probe head is one-dimensional, followed by a sigmoid function to normalize the output to within range $[0, 1]$. We train the probe head using binary cross-entropy loss.

---

[1]If not stated otherwise, $k = 10$.

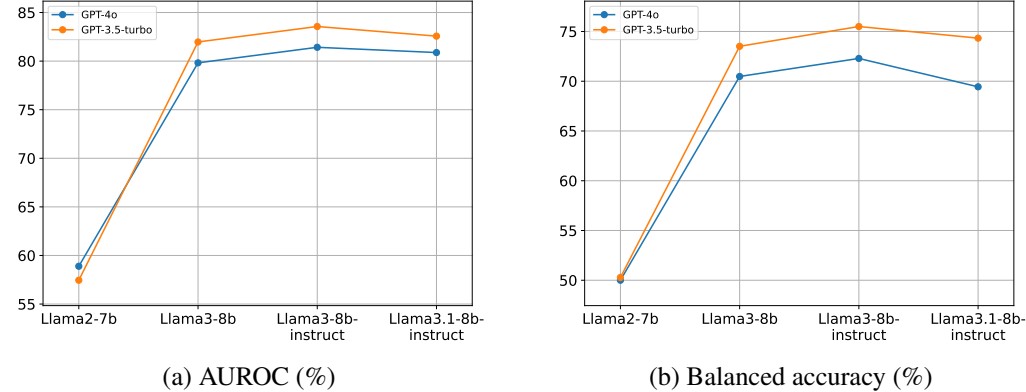

(a) AUROC (%)                    (b) Balanced accuracy (%)

Figure 1: Evaluation results on TriviaQA with different $f_E$. As the size and capability of $f_E$ increases, there is a clear improvement in AUROC. GPT-3.5-turbo is consistently better learned than GPT-4o.

**Supervised fine-tuning using LoRA.**  Empirical evidence suggests that finetuning can often improve performance over fixed representations (He et al., 2021). We fine-tune $f_E$ using Low-rank Adaption (LoRA) following Hu et al. (2021). In this setup, we train a single linear layer with a sigmoid activation as the probe head and apply LoRA to all layers preceding the final probe.

**Evaluation metrics.**  We evaluate the performance of our evaluators using a range of metrics. Since $y$ represents a probability, we discretize both $y_{\text{true}}$ and $y_{\text{pred}}$ (classifying based on whether $y \geq 0.5$) to compute metrics including accuracy and F1 score. For metrics like AUROC and AUPRC, we keep $y_{\text{pred}}$ as a probability, discretizing only $y_{\text{true}}$ to treat it as a classification task. To address data imbalance, we subsample the data to create a class-balanced test set, which we use to compute balanced accuracy. Additionally, we compute selective metrics to assess whether the model is well-calibrated to reject the uncertain answers. Unless stated otherwise, all metrics in the plots are presented as percentages.

## 4 Results

We constructed two datasets from TriviaQA and MMLU. For $f_G$, we collected answers from GPT-3.5-turbo and GPT-4o. For $f_E$, we used Llama2-7b, Llama3-8b, Llama3-8b-instruct, and Llama3.1-8b-instruct as the evaluator backbone to obtain representations.

### 4.1 Dataset TriviaQA

The TriviaQA dataset contains a wide range of trivia questions and corresponding keyword lists of answers. The first trend we observe is that scaling up the Evaluator backbone improves performance. With Llama2-7b as $f_E$'s backbone, the AUROC for predicting the correctness of GPT-4o on TriviaQA is 58.89%. When scaled up to Llama3-8b, the AUROC increases significantly to 79.82%, as shown in Figure 1(a) – a notable improvement compared with the 14.29% increase in $f_E$'s parameters.

The training method of the "backbone" LLM contributes marginally to performance improvement in uncertainty estimation. The AUROC for GPT-4o prediction increases to 81.42% when Llama3-8b-instruct, which is fine-tuned via instruction (Wei et al., 2022), is used as $f_E$. This trend, demonstrated in Figure 1, shows that AUROC consistently improves as the backbone of $f_E$ becomes more intelligent. The same pattern is observed across balanced accuracy, AUPRC, F1 score, and other metrics. Full results can be found in Appendix A.

Fine-tuning $f_E$ using LoRA enhances performance compared to only training the probe head. With the best-performing Llama3-8b-instruct, we achieved an AUROC of 82.16% and balanced accuracy of 72.09% for predicting correctness of GPT-4o. Fine-tuning the entire $f_E$ refines the representation, enabling more accurate uncertainty predictions.

Tables 1, 2 and 5 in the appendix provide a comprehensive set of evaluations on TriviaQA. For GPT-4o, the top evaluator achieves over 90% in both AUPRC and F1 score. The LoRA-fine-tuned evaluator is well-calibrated, with selective accuracy and selective F1 increasing, indicating reduced

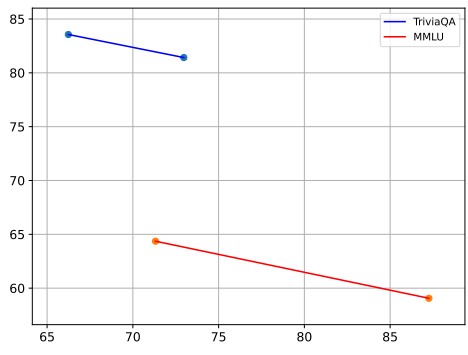 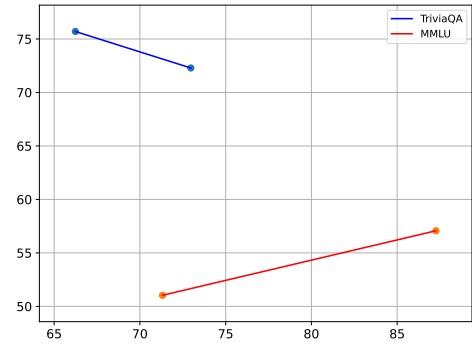

(a) Relationship between Generator accuracy (%) and probe head AUROC (%)

(b) Relationship between Generator accuracy (%) and probe head balanced accuracy (%)

Figure 2: GPT-3.5-turbo achieved 66.22% accuracy on TriviaQA and GPT-4o achieved 72.97%; GPT-3.5-turbo achieved 71.32% accuracy on MMLU and GPT-4o achieved 87.26%. As the accuracy of Generator increases, AUROC of Evaluator decreases monotonically. This negative trend is observed on both TriviaQA and MMLU datasets. As for the balanced accuracy on MMLU, the positive trend can be attributed to the imbalance between class 0 and class 1.

prediction uncertainty as model confidence grows. For GPT-3.5-turbo, the best evaluator achieves an AUROC of 84.69%, accuracy of 82.16%, and balanced accuracy of 76.53%, with AUPRC and F1 in the high 90%. $f_E$ for GPT-3.5-turbo is more calibrated than $f_E$ for gpt-4o, as evidenced by its higher selective AUROC. Our hypothesis is that GPT-3.5-turbo's size and capability are closer to the Llama3 families, resulting in more aligned representations and improved calibrations. More results on selective performance are presented in Figure 3.

## 4.2 Dataset MMLU

However, $f_E$'s performance varies depending on the task. When using the MMLU dataset (Hendrycks et al., 2021), which emphasizes reasoning, the Evaluator struggles to capture the uncertainty. For GPT-4o, the balanced accuracy drops to 52.3% and AUROC to 59.06%, despite the high accuracy of 85.15%. This drop is likely due to GPT-4o's strong performance on MMLU, with an answer accuracy of 87.26%, causing a highly imbalanced training set. This data imbalance is further reflected in the correlation between Generator accuracy and Evaluator performance, as shown in Figure 2. Predicting GPT-3.5-turbo performs better, with a balanced accuracy of 56.04% and AUROC of 64.36%. Fine-tuning with LoRA provides limited gains, likely due to: 1) the nature of the questions, which, unlike TriviaQA, are not explicitly tied to the answers, and 2) the multiple-choice format, which includes four answer choices, leading to incorrect options often containing irrelevant information. These factors complicate the representation learning. Full MMLU results are provided in Tables 3 to 5.

## 5 Discussion and Conclusions

In this paper, we explored the question to what extent the question-answering accuracy of a stronger LLM can be predicted by a weaker LLM. We found that, using a stronger model's predictive uncertainty to learn an evaluator parameterized by a significantly smaller model, it is in fact possible to predict a stronger model's ability to provide a correct answer. We find that the evaluators trained on responses from stronger models also well-calibrated: the predicted probabilities they output closely mimic the true probabilities of generators being correct in answering a question. In fact, we believe model-specific features are learned by these evaluators. (See Appendix C for more details). Our results are important both for the engineering of safe LLMs, in that they guide developers of these models, as well as for effective technical AI governance, as they give end users of LLMs ways to ascertain the accuracy of the model they use, even when these are black-boxes. We believe that further exploration of the relationship between weaker evaluators and stronger generators, such as whether self-evaluation (Kadavath et al., 2022) performs better than external evaluation, and whether evaluators are learning features specific to different generators, is important towards building more interpretable frontier models.

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

# Appendix

## Appendix A    Further Experimental Results

Note: All results in the tables and the plots below are of percentage.

Table 1: Probe Head Evaluation Results on TriviaQA: $f_G$ = GPT-3.5-turbo.

| $f_E$ Metrics | Llama3.1-8b-instruct | Llama3-8b-instruct | Llama3-8b | Llama2-7b |
|---|---|---|---|---|
| AUROC | 82.57 | 83.56 | 81.97 | 57.45 |
| Accuracy | 81.70 | 79.12 | 79.70 | 73.70 |
| balanced accuracy | 91.63 | 75.50 | 73.51 | 50.28 |
| AUPRC | 88.08 | 92.68 | 91.73 | 78.32 |
| F1 Score | 87.48 | 85.42 | 86.29 | 84.79 |
| Selective Accuracy | 90.60 | 87.87 | 86.89 | 76.60 |
| Selective AUROC | 81.06 | 89.07 | 85.48 | 52.38 |
| Selective F1 | 94.04 | 91.43 | 91.22 | 86.34 |

Table 2: Probe Head Evaluation Results on TriviaQA: $f_G$ = GPT-4o.

| $f_E$ Metrics | Llama3.1-8b-instruct | Llama3-8b-instruct | Llama3-8b | Llama2-7b |
|---|---|---|---|---|
| AUROC | 80.88 | 81.42 | 79.82 | 58.89 |
| Accuracy | 83.95 | 81.79 | 81.42 | 81.10 |
| Balanced Accuracy | 69.44 | 72.29 | 70.48 | 50.00 |
| AUPRC | 94.06 | 94.37 | 83.55 | 84.06 |
| F1 Score | 90.82 | 88.65 | 88.51 | 89.56 |
| Selective Accuracy | 92.39 | 91.79 | 91.06 | 83.98 |
| Selective AUROC | 71.23 | 72.82 | 72.51 | 55.93 |
| Selective F1 | 95.43 | 94.94 | 94.48 | 90.85 |

Table 3: Probe Head Evaluation Results on MMLU: $f_G$ = GPT-3.5-turbo.

| $f_E$
Metrics | Llama3-8b-instruct | Llama3-8b | Llama2-7b |
|---|---|---|---|
| AUROC | 64.11 | 63.80 | 65.28 |
| Accuracy | 70.07 | 70.23 | 69.26 |
| Balanced Accuracy | 57.07 | 55.86 | 55.31 |
| AUPRC | 78.62 | 77.83 | 79.28 |
| F1 Score | 80.99 | 81.33 | 80.70 |
| Selective Accuracy | 76.53 | 76.30 | 77.50 |
| Selective AUROC | 58.48 | 56.06 | 57.83 |
| Selective F1 | 85.84 | 85.68 | 86.74 |

Table 4: Probe Head Evaluation Results on MMLU: $f_G$ = GPT-4o.

| $f_E$
Metrics | Llama3-8b-instruct | Llama3-8b | Llama2-7b |
|---|---|---|---|
| AUROC | 55.65 | 59.96 | 61.89 |
| Accuracy | 85.36 | 85.68 | 85.61 |
| Balanced Accuracy | 50.98 | 51.04 | 50.83 |
| AUPRC | 87.97 | 89.05 | 89.83 |
| F1 Score | 92.11 | 92.31 | 88.81 |
| Selective Accuracy | 87.00 | 88.07 | 83.98 |
| Selective AUROC | 53.91 | 53.65 | 54.76 |
| Selective F1 | 92.60 | 92.83 | 93.61 |

Table 5: LoRA Evaluation Results: $f_E$ = Llama3-8b-instruct.

| $\mathcal{D}$
$f_G$
Metrics | TriviaQA | | MMLU | |
|---|---|---|---|---|
| | gpt-3.5-turbo | gpt-4o | gpt-3.5-turbo | gpt-4o |
| AUROC | 84.69 | 82.16 | 64.36 | 59.06 |
| Accuracy | 82.16 | 84.43 | 67.93 | 85.15 |
| Balanced Accuracy | 76.53 | 72.09 | 56.04 | 52.38 |
| AUPRC | 92.87 | 94.35 | 78.83 | 89.02 |
| F1 Score | 88.38 | 90.73 | 79.50 | 91.66 |
| Selective Accuracy | 91.14 | 92.50 | 76.48 | 87.82 |
| Selective AUROC | 89.30 | 81.24 | 59.86 | 55.75 |
| Selective F1 | 94.32 | 95.39 | 62.34 | 56.11 |

 # Appendix B    Visualization of Additional Evaluation Metrics

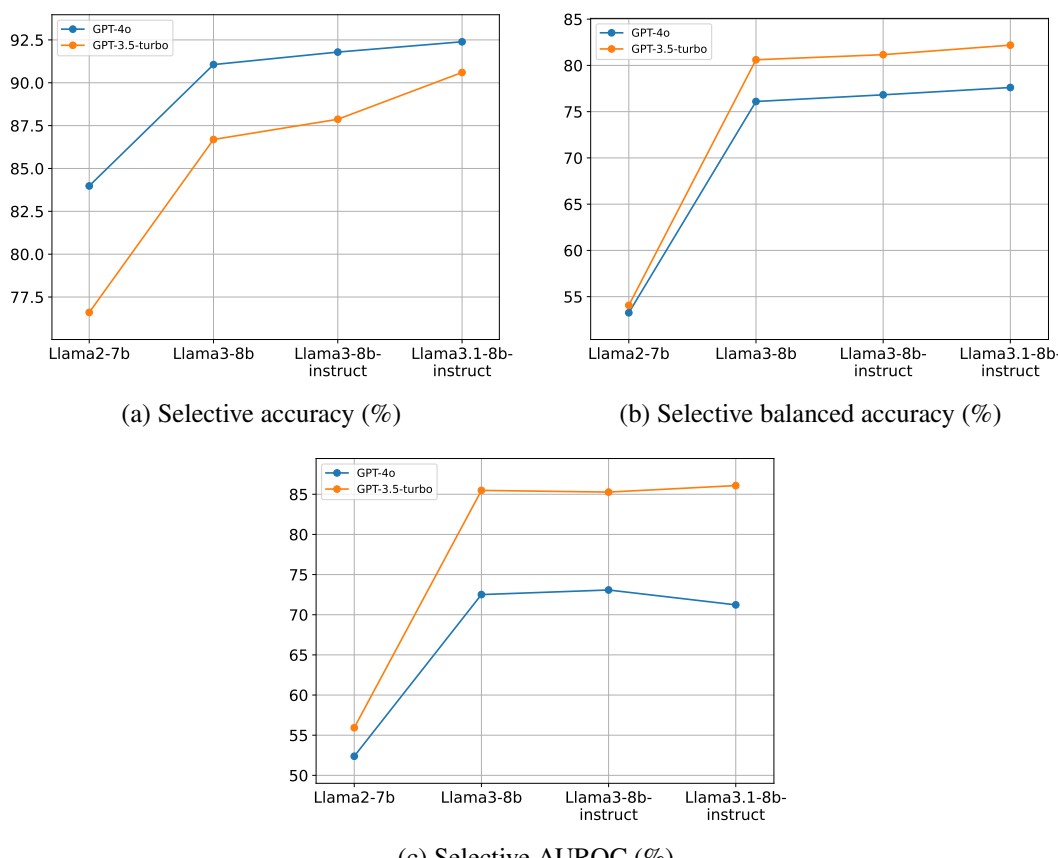

(a) Selective accuracy (%)              (b) Selective balanced accuracy (%)

(c) Selective AUROC (%)

Figure 3: Selective prediction on TriviaQA.

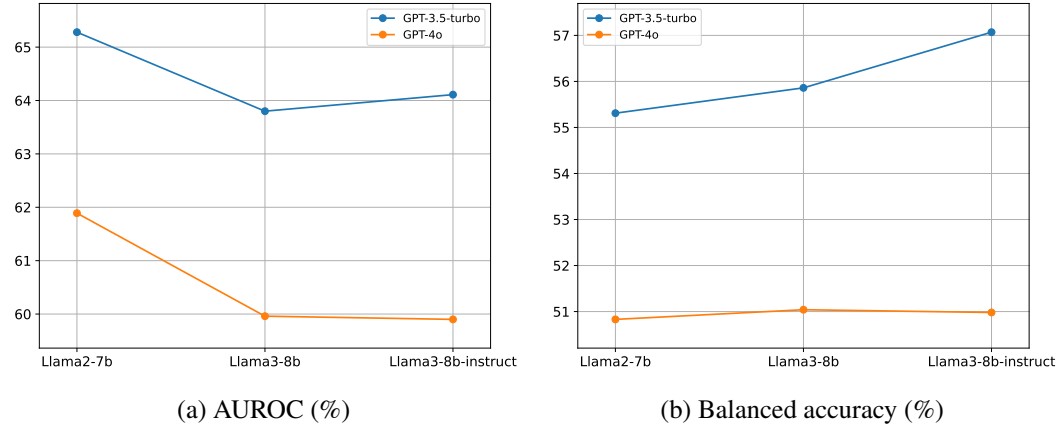

(a) AUROC (%)                    (b) Balanced accuracy (%)

Figure 4: Evaluation results on MMLU with different $f_E$.

# Appendix C    Further study: Are evaluators learning generator-specific features?

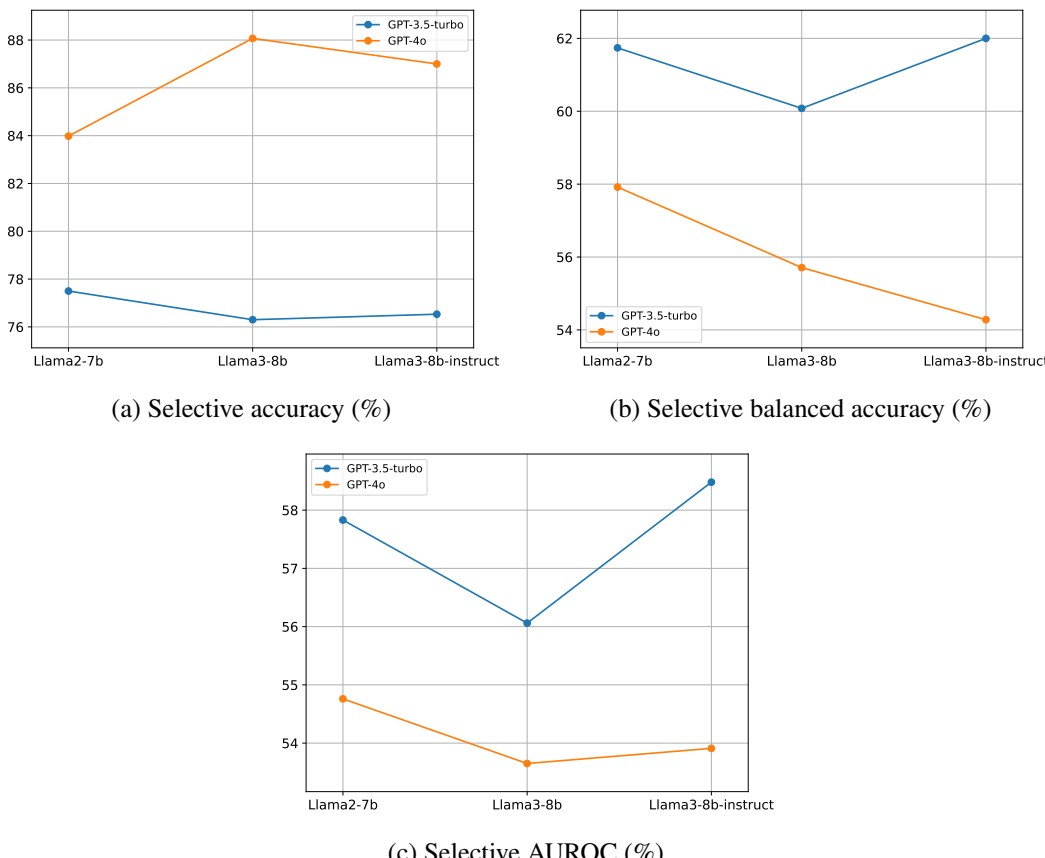

(a) Selective accuracy (%)

(b) Selective balanced accuracy (%)

(c) Selective AUROC (%)

Figure 5: Selective prediction on MMLU.

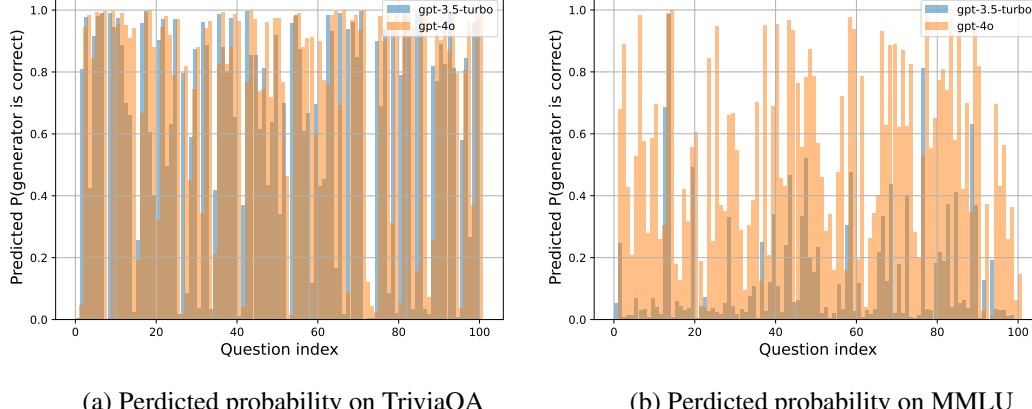

(a) Perdicted probability on TriviaQA

(b) Perdicted probability on MMLU

Figure 6: We visualize the predicted probability of $f_E$ for different $f_G$ on the first 100 questions of the test set. For TriviaQA plot we are use llama3-8b and for MMLU llama3-8b-instruct as the evaluator. For both datasets we observe that the evaluator learns different distributions from different generators. This indicates that the same evaluator can learn generator-specific features, leading to the different predictve distribution of $\mathbf{P}(f_g$ is correct$)$. For TriviaQA, we run a Kolmogorov–Smirnov test and get $K = 0.21$, p-value$= 0.02$; for MMLU, $K = 0.65$, p-value$= 3.1e−20$. The low p-values can make us reject the null hypothesis and demonstrate that the distributions are different.

