# OpenReview forum: "Weak-to-Strong Confidence Prediction"
_NeurIPS.cc/2024/Workshop/SafeGenAi — SafeGenAi Poster_

### Official Review · Reviewer_c9Ja · 2024-10-08
**The paper proposed an evaluator agent to judge the outcomes of another agent. The main claim is that a small evaluator can achieve high performance.**

**Rating:** 3
**Confidence:** 4

**Review:**

The idea of providing safety guarantees via a gatekeeper model is interesting and promising.

However, the paper in its current form precludes acceptance. The three claims stated in the introduction are weakly supported across the paper. For instance, regarding calibration, lines 124-125 only provide speculation as evidence. The results are not appropiately explained. Following the previous example, lines 125-126, force the reader to interpret the relevance of Figure 3. How does Figure 3 complement Figure 1? This has to be explicit in order to strengthen the results of the paper.

I encourage the authors to continue improving this paper in order to make an impactful contribution to the field.

---

### Official Review · Reviewer_tACv · 2024-10-09
**Review of "Weak-to-Strong Confidence Prediction"**

**Rating:** 3
**Confidence:** 4

**Review:**

### **Summary**

The paper investigates whether the accuracy of large language models (LLMs) in generating factual responses can be reliably predicted by smaller, weaker models. The authors propose using a secondary model (the "evaluator") to estimate the confidence of answers produced by a primary model (the "generator").

### **Strengths**

1. The paper addresses a practical problem.

### **Weaknesses**

1. The paper is not well-written, with numerous grammatical errors that make it difficult to comprehend.

2. The approach of using a probe to analyze a model's internal state and detect accuracy is quite common. The paper does not clarify its unique contribution. Is the novelty in using a small LLM instead of a simple classifier, or is it directly predicting confidence scores?

3.  The statement in the abstract:"existing methods largely fail to provide reliable uncertainty estimates in practice, and the lack of comparability across methods makes measuring progress difficult, necessitating the development of more robust methods", is not adequately supported or resolved in the paper.

4. The paper does not explain why smaller models might estimate the confidence of larger models more effectively or why directly using the larger model’s own verbal or probabilistic output for confidence estimation is inadequate.

5. The experimental setup is insufficient. The correlation between the generator and evaluator models is visualized only on two datasets with limited base models and metrics, which is not enough to support the conclusions. Additionally, there is no comparison with existing state-of-the-art methods for confidence estimation.

---

### Official Review · Reviewer_GxGb · 2024-10-09
**Review for the Paper: "Weak-to-Strong Confidence Prediction"**

**Rating:** 5
**Confidence:** 3

**Review:**

### **Summary**

This paper explores the concept of weak-to-strong confidence prediction in large language models (LLMs) by introducing a novel framework where smaller, weaker LLMs evaluate the accuracy and uncertainty of stronger, larger models. The idea centers around using an evaluator LLM to predict the probability of the correctness of a generated answer from a larger model (the generator LLM). This approach provides valuable insights into uncertainty quantification, particularly in detecting hallucinations in black-box LLMs. The paper presents experimental results from two datasets, TriviaQA and MMLU, demonstrating how well smaller models can predict the accuracy of more capable models, such as GPT-3.5-turbo and GPT-4

### **Strengths**

1. **Practical Contributions**: The idea that the evaluator model can operate locally, without needing access to the larger model's internal mechanics, is highly practical for applications where privacy or server access is restricted. This has significant implications for LLM deployment in sensitive contexts
2. **Strong Results on Multiple Metrics**: The paper evaluates performance using a wide range of metrics, such as AUROC, F1 score, accuracy, and selective accuracy, which offers a thorough and balanced view of the system's performance. The gains in calibration across different model configurations (with LoRA fine-tuning, probe heads, etc.) further solidify the proposed method’s robustness

### **Weaknesses**

1. **Limited Scope of Evaluation**: The paper focuses on two datasets (TriviaQA and MMLU) and mainly on factual question-answering tasks. It would be valuable to see whether this weak-to-strong confidence prediction framework generalizes to other tasks, such as open-ended text generation or dialogue
2. **Data Imbalance Issues**: The paper reports a notable drop in performance on the MMLU dataset, attributed to data imbalance and the strong performance of GPT-4. However, the strategies for addressing this imbalance could be better elaborated. Additionally, the discussion could include more insights into how to handle such limitations in practice
3. **Task-Specific Results**: The performance of the evaluator models depends heavily on the task at hand. For example, the framework performs well on TriviaQA but struggles with MMLU. This variability suggests that the generalizability of the approach across different types of tasks (factual recall versus reasoning) might need further investigation

### **Detailed Comments**

- The authors thoroughly evaluate the performance using a wide range of metrics, including AUROC, accuracy, balanced accuracy, and selective accuracy. For example, the best-performing Llama3-8b-instruct achieved an AUROC of 82.16% on GPT-4’s predictions in the TriviaQA dataset. These results provide a clear understanding of the framework's strengths and weaknesses, especially regarding calibration and confidence prediction
- The lower performance on the MMLU dataset (with balanced accuracy dropping to around 50% for GPT-4 predictions) suggests that the evaluator model struggles with more reasoning-based tasks. The paper attributes this to both data imbalance and task complexity, but future work could explore more advanced techniques to handle such issues or improve evaluator calibration for complex, reasoning-intensive datasets
- The paper provides evidence that fine-tuning the evaluator model with LoRA significantly improves performance across various metrics. However, more discussion on how LoRA influences the learned representations and how this differs from simple probe-head training would add depth to the paper's claims about fine-tuning benefits